applied mathematics/fluid mechanics

microswimmer control, flow control, robustness, controllability

**Author for correspondence:**
Benjamin J. Walker
e-mail: benjamin.walker@maths.ox.ac.uk

# Control and controllability of microswimmers by a shearing flow

Clément Moreau[1], Kenta Ishimoto[1], Eamonn A. Gaffney[2] and Benjamin J. Walker[2]

[1]Research Institute for Mathematical Sciences, Kyoto University, Kyoto, 606-8502, Japan
[2]Wolfson Centre for Mathematical Biology, Mathematical Institute, University of Oxford, Oxford OX2 6GG, UK

CM, 0000-0002-8557-1149; KI, 0000-0003-1900-7643; BJW, 0000-0003-0853-267X

With the continuing rapid development of artificial microrobots and active particles, questions of microswimmer guidance and control are becoming ever more relevant and prevalent. In both the applications and theoretical study of such microscale swimmers, control is often mediated by an engineered property of the swimmer, such as in the case of magnetically propelled microrobots. In this work, we will consider a modality of control that is applicable in more generality, effecting guidance via modulation of a background fluid flow. Here, considering a model swimmer in a commonplace flow and simple geometry, we analyse and subsequently establish the efficacy of flow-mediated microswimmer positional control, later touching upon a question of optimal control. Moving beyond idealized notions of controllability and towards considerations of practical utility, we then evaluate the robustness of this control modality to sources of variation that may be present in applications, examining in particular the effects of measurement inaccuracy and rotational noise. This exploration gives rise to a number of cautionary observations, which, overall, demonstrate the need for the careful assessment of both policy and behavioural robustness when designing control schemes for use in practice.

## 1. Introduction

Synthetic microscale swimmers are becoming increasingly prevalent, with the canonical biological examples of spermatozoa and bacteria informing and motivating the development of modern artificial microrobots and active particles. A natural question that arises is that of guidance and control, particularly for engineered swimmers with targeted applications. Examples to date have included the navigation of an intricate environment or

microdevice [1,2], the delivery of therapeutic cargo for medical interventions [3–5] or hydrodynamically disguising the motions of another swimmer [6]. In particular, as continued developments in fabrication and biohybridization advance the potential utility of microswimmers, the task of directing a microswimmer along a path or towards a desired goal becomes ever-more pertinent.

However, realizing the seemingly simple objectives of guidance and control can be complex. Much of this complexity is determined by the modality of swimmer control or propulsion, which can range from magnetically propelled microrobots to acoustically driven synthetic swimmers [7,8]. As such, there are many different strategies and modalities for realizing swimmer control. Magnetotactic microswimmers, such as the rotating magnetic helices that are the subject of many recent studies [9–12], naturally admit simple control via modulation of the applied magnetic field. Further examples include the recent work of Khalil *et al.* [13], which exploits the magnetism-driven rotation of a microrobot to drive fluid flow, in turn directing passive microbeads. Other approaches consider swimmers as internally controlled or 'smart agents', capable of effecting changes in their own direction or speed in response to their environment. These self-regulated swimmers, often coupled with adaptive or learned strategies [2,6,14], are able to navigate changing multiswimmer environments while achieving sophisticated goals, for example, the hydrodynamic cloaking of another swimmer through maintenance of particular relative positioning [6]. However, methods tailored to guiding magnetic or smart-sensing swimmers are inherently restricted to these subclasses of microswimmer, with neither of these approaches being suitable for controlling the movement of simpler, non-magnetic, synthetic swimmers or biological cells, such as bacteria and spermatozoa.

A mechanism that avoids these restrictions is that of flow control, in which the guidance of a generic microswimmer is realized via the modulation of a generated fluid flow. Indeed, the field of microfluidics has advanced rapidly over the past two decades, with finely calibrated changes to fluid flows now able to be effected on time scales of milliseconds [15]. While this method of control is naturally restricted to contexts in which flow can be applied, it is otherwise universally applicable to microswimmers, in particular not requiring on-board sensors or magnetic components. Despite this broad applicability, this flow-based approach has, to the best of our knowledge, only briefly been explored [13,16]. However, removed from the context of control, the general interactions of swimmers with background flows have been extensively considered, both experimentally and theoretically [17–22], with observations including the remarkable upstream rheotaxis of spermatozoa when confined [23–25]. Such studies build upon classical works of the twentieth century for the dynamics of bodies in flow, with perhaps the best known being the study of Jeffery [26] on the intricate orbits of passive ellipsoids in a constant shear flow, to which we direct the interested reader for an exploration of these complex behaviours. Recently, it has been noted that Jeffery's analytical results can be applied in more generality, with Jeffery's orbits accurately representing the time-averaged dynamics of both a flagellated swimmer and bodies with particular symmetries [16,27]. Thus, the consideration of the motion of an ellipsoid in flow presents itself as a viable first avenue for the exploration of the guidance of more complex swimmers. Therefore, with the governing equations of such an idealized swimmer model being sufficiently simple so as to be amenable to analysis and ready exploration, the first objective of this work will be to formulate the control system describing a flow-guided ellipsoid, from which we will formally evaluate the theoretical controllability properties of the swimmer.

Controllability addresses the theoretical existence of controlled trajectories between given initial and target states of a system. Once this existence is established, one can seek to find the best trajectory with respect to the minimization of a certain objective function, such as time, using optimal control theory. The first theoretical studies of microswimmer control, as well as many subsequent works, considered swimmers that were set in motion by their own deformation, whether for a general deformable body [28–32], or a slender robot comprising rigid links connected by elastic joints [33–36]. This latter type of swimmer has also been more recently analysed when driven by an external magnetic field, from the points of view of both controllability [37–39] and optimal control [40]. Further to laying a formal mathematical foundation that fosters understanding of microswimming at a theoretical level, the control-theory standpoint can provide both guidance for designing microdevices and an optimal way of controlling them. The present analysis will therefore seek to establish the viability of flow control as a means of guidance in the simple but broadly applicable scenario considered in this work, while motivating and informing future considerations of complex fluid and swimmer control problems.

While theoretical results of swimmer controllability would serve to address the abstract and idealized problem of flow-mediated control, it will only go so far in evaluating more practical degrees of suitability. In particular, in order for any method of control to be of practical efficacy, a certain level of robustness must be established. For instance, any approach that prescribes a policy of flow control to elicit a

particular swimmer behaviour should be robust to inaccuracies in the measured geometry of the swimmer, with the theory of Jeffery's orbits highlighting the impacts of shape on the response of even simple bodies to simple flows. Another significant factor in many microscale processes is diffusive noise, such as the rotational diffusion of bacterial swimmers that invariably leads to non-straight swimming paths in a quiescent environment. This desired robustness is naturally present in feedback-control mechanisms of guidance, such as smart and programmable agents [2,6,14,41], but is not guaranteed in the broader context of microswimmer control. Hence, as the second objective of this study, we will investigate the robustness of a range of both designed and optimized flow control policies to variations in swimmer shape, as might be particularly pertinent when considering a heterogeneous population, in addition to the interactions between flow and diffusive effects.

Hence, we will proceed by formulating the simple control system that captures the planar dynamics of an idealized swimmer in a background shear flow. Applying well-known tools of control theory, we will analytically interrogate the local controllability around trajectories of the system and the problem of minimal-time control, additionally providing numerical exemplars to illustrate the key features of the coupled flow-control system. With these theoretical explorations in hand, we will move to evaluate the impacts of shape on the design of control policies, including the resulting complex range of behaviours exhibited by an immersed swimmer. Further, we will incorporate the effects of rotational diffusion into our swimmer model, exploring the relation between applied flow and the measured diffusivity of a swimmer by explicit example. In doing so, we will seek to evaluate the plausibility of flow-mediated guidance as a mechanism for both theoretical and practical microswimmer control.

## 2. A simple flow-driven control system

We will consider the planar motion of an active prolate spheroid in a background Newtonian shear flow at zero Reynolds number, with this idealized axisymmetric swimmer being a common model for a range of biological and artificial microswimmers, applicable even to the beat-averaged dynamics of a flagellated swimmer or non-axisymmetric bodies [16,27]. We parametrize its shape via the aspect ratio $r \geq 1$, the dimensionless ratio of its major to minor axes, with a sphere corresponding to $r = 1$ and an elongated ellipsoid to $r > 1$. In more generality, this may be replaced with an effective aspect ratio where justified, with the analysis that follows otherwise unchanged. As an explicit example, Walker *et al.* [16] calculated an effective aspect ratio of $r \approx 5.49$ for a model *Leishmania mexicana* in shear flow, a flagellated microswimmer that, despite periodic yawing due to its flagellar motion, can be well represented by the simple spheroid model. Written with respect to the laboratory frame, with the plane containing the swimmer and its major axis being described by coordinates $(x, y)$ in the directions of orthogonal unit vectors $e_x$, $e_y$, respectively, we write the background shear flow simply as

$$u(x, y, t) = \gamma(t)y e_x, \tag{2.1}$$

where $t$ is dimensionless time and $\gamma(t)$ is the time-dependent shear rate, which we will later prescribe and refer to as the *control*. Note that this flow is consistent with the assumed planar motion of the swimmer, illustrated in figure 1. In practice, a broad range of shear rates have long been realizable in microdevices [42], spanning regimes of likely utility for control. For example, taking $r = 5.49$ to represent *Leishmania mexicana* as noted above, a nominal shear rate of $\gamma = 100\,\text{s}^{-1}$ is able to fully rotate a model cell approximately three times per second, with lower rates accordingly generating slower rotations. In appendix A, we briefly illustrate, in dimensional variables and with reference to swimmer length scales, the control of *Leishmania mexicana* at lower shear rates and additionally explore the effects of microfluidic response times on the resulting dynamics.

Assuming further that the heading of the swimmer due to its own activity is both fixed in the body and parallel to its major axis, we parametrize the orientation of the swimmer in the plane by an angle $\theta$, where taking $\theta = 0$ corresponds to orienting the swimmer along the positive $e_x$ direction. Following [43], the familiar equations of motion for this swimmer may be concisely written as

$$\begin{cases} \dot{x} = \cos\theta + \gamma(t)y, \\ \dot{y} = \sin\theta, \\ \dot{\theta} = \frac{1}{2}\gamma(t)(-1 + E\cos 2\theta), \end{cases} \tag{2.2}$$

where a dot denotes a derivative with respect to time. Here, $E = (r^2 - 1)/(r^2 + 1)$ represents the effects of swimmer shape on the motion, from which it is clear that $r = 1$ gives the orientation-independent

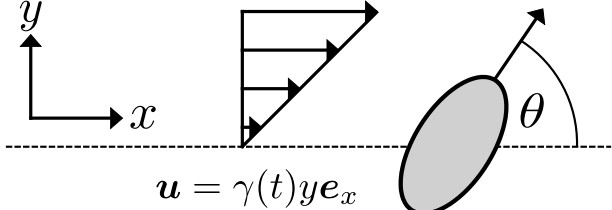

**Figure 1.** A swimmer in flow. We model the swimmer as a prolate spheroid with propulsive velocity in the direction indicated by the attached arrow, which makes an angle $\theta$ to the $x$-axis of the laboratory frame. The swimmer moves in the $xy$-plane, acted on by the shear flow $\boldsymbol{u}$, which acts along the $\boldsymbol{e}_x$ direction.

dynamics of a sphere. A brief derivation of these equations is given in appendix B, and we remark that the evolution of $\theta$ is that of a planar Jeffery's orbit if $\gamma(t)$ is held constant [26]. With the swimmer position and orientation being represented by the tuple $z = [x, y, \theta]^{\mathrm{T}}$, we will later refer to this control system simply as $\dot{z} = f(z, \gamma)$, where $f$ encodes the relations of equation (2.2). We note that we have assumed that the propulsive velocity of the swimmer is of unit magnitude, which, subject to simple rescalings of time and spatial coordinates, retains the generality of a swimmer with a fixed but otherwise arbitrary speed. This is equivalent to non-dimensionalizing velocities via the fixed linear speed of the swimmer, with the associated length and time scales otherwise arbitrary so long as the assumption of zero Reynolds number remains appropriate, as discussed further in appendix A. In writing this deterministic system, we are assuming that the effects of translational and rotational noise are subdominant to those of self-propulsion and flow, valid for swimmers such as spermatozoa and *Leishmania mexicana*, though we will later modify this system to include rotationally diffusive effects, which are well known to be significant in the dynamics of some motile bacteria [44].

In what follows, we will consider controls that are prescribed as well as those that are the solution to a stated optimal control problem, typically seeking to optimize travel time given suitable constraints that we will later formulate in detail. To solve such optimization problems, we will make use of the CasADi framework [45], subsequently using in-built ordinary differential equation solvers in MATLAB® for the repeated solution of the deterministic control system (2.2) [46].

# 3. Controllability analysis

Consider a constant shear rate $\bar{\gamma}$, which we will call the *reference control*. Given an initial state $\bar{z}_0 = [\bar{x}_0, \bar{y}_0, \bar{\theta}_0]^{\mathrm{T}}$ for the swimmer, let $\bar{z} = [\bar{x}, \bar{y}, \bar{\theta}]^{\mathrm{T}}$ be the solution of system (2.2) starting at $\bar{z}_0$ with the constant shear rate $\bar{\gamma}$, on the time interval $[0, T]$; we will call $\bar{z}$ the *reference trajectory*. In this section, we address the controllability properties of the flow-driven swimmer in a local sense around the reference trajectory $\bar{z}$, trying to answer the following question, which constitutes an important theoretical feature for a controlled system: do 'small' variations of the reference control $\bar{\gamma}$ allow one to target any position and orientation within a neighbourhood of the reference trajectory? A particular situation of interest is the case of $\bar{\gamma} = 0$, in which we seek to control the position and orientation of the swimmer by generating a shear flow of 'small' amplitude.

The formal definition of this notion can be stated as follows [47]: system (2.2) is said to be locally controllable around the trajectory $(\bar{z}, \bar{\gamma})$ if, for all $\varepsilon > 0$, there exists neighbourhoods $\mathcal{V}$ and $\mathcal{W}$ of $\bar{z}_0$ and $\bar{z}(T)$ such that, for all $z_0$ in $\mathcal{V}$ and $z_1$ in $\mathcal{W}$, there exists a bounded control $\gamma$ defined on the time interval $[0, T]$, satisfying $|\gamma(t) - \bar{\gamma}| \leq \varepsilon$ for all $t \in [0, T]$, and such that the solution of system (2.2) starting at $z_0$ satisfies $z(T) = z_1$. Figure 2 illustrates this notion of local controllability.

To study the local controllability of system (2.2), we consider the linearized control system of system (2.2) around $(\bar{z}, \bar{\gamma})$, which reads

$$\dot{z} = A(t)z + B(t)\gamma(t), \tag{3.1}$$

with $A = (\partial f(z, \gamma)/\partial z)(\bar{z}, \bar{\gamma})$ and $B = (\partial f(z, \gamma)/\partial \gamma)(\bar{z}, \bar{\gamma})$. It is well known (see for instance [47, theorem 3.6]) that the nonlinear system is locally controllable around $(\bar{z}, \bar{\gamma})$ if the linearized system (3.1) is (globally) controllable. To check if this is the case, we can apply a simple algebraic 'Kalman-type' condition for time-varying linear control systems [48,49]: let $B_1(t) = \dot{B} - AB$ and $B_2(t) = \dot{B}_1 - AB_1$; then system (3.1) is controllable in time $T$ if and only if there exists $t \in [0, T]$ such that the matrix

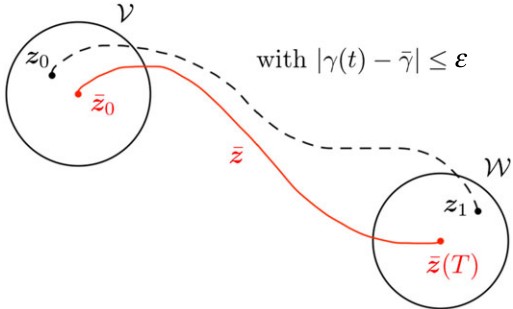

**Figure 2.** Schematic of local controllability around a reference trajectory $\bar{z}$, in which the swimmer moves along the dashed trajectory between the neighbourhoods $\mathcal{V}$ and $\mathcal{W}$ while the control $\gamma(t)$ is kept close to the reference control $\bar{\gamma}$. Informally, the system is locally controllable around the trajectory $\bar{z}$ if such a dashed trajectory joining $z_0$ and $z_1$ exists for all states $z_0 \in \mathcal{V}$ and $z_1 \in \mathcal{W}$.

$(B(t) | B_1(t) | B_2(t))$ is invertible. The determinant $\Delta$ of $(B(t) | B_1(t) | B_2(t))$ can be obtained through straightforward calculations:

$$\Delta = \frac{1}{32}\,\bar{\gamma}(E\cos 2\bar{\theta} - 1)[2\bar{\gamma}\bar{y}E^2(\cos 4\bar{\theta} + 4\cos\bar{\theta} - 1) + (E - 1)\{4 + 2\cos\bar{\theta} + E(1 + \cos\bar{\theta})\}].$$

Let us first assume that the reference control is null ($\bar{\gamma} = 0$). It is worth noticing that, in this case, the reference trajectory explicitly reads $\bar{z}(t) = [\bar{x}_0 + t\cos\bar{\theta}_0, \bar{y}_0 + t\sin\bar{\theta}_0, \bar{\theta}_0]^{\mathrm{T}}$, i.e. the swimmer goes in a straight line in the direction given by the initial angle $\bar{\theta}_0$. Moreover, we immediately see that $\Delta = 0$. Hence, the linearized system is not controllable. Although the fact that the linearized system is not controllable does not mean that the swimmer is not locally controllable around the straight reference trajectory $\bar{z}(t)$, it does suggest that some directions in the state space are difficult to attain in a neighbourhood of the reference trajectory by using a variable shear rate of small amplitude.

Alternatively, we may seek to control only the position $(x, y)$, rather than both the position and orientation $(x, y, \theta)$. For this purpose, we define *partial local controllability* as in [50,51] by restricting controllability requirements to the variables $x$ and $y$. A similar analysis on the linearized system can be conducted to evaluate the local partial controllability properties of the system projected onto the $xy$-plane [51]. For $\bar{\gamma} = 0$, the determinant of the partial Kalman matrix, defined in the same manner as above, is equal to

$$\frac{1}{2}\,(\bar{y}_0 + t\sin\bar{\theta}_0)\cos\bar{\theta}_0(E\cos 2\bar{\theta}_0 - 1). \tag{3.2}$$

We readily conclude that, if $\bar{\gamma} = 0$, the swimmer is partially locally controllable around the trajectory $\bar{z}$ for the position variables $x$, $y$, unless (i) $\cos\bar{\theta}_0 = 0$ or (ii) both $\sin\bar{\theta}_0 = 0$ and $\bar{y}_0 = 0$.

We now turn to the more general case with $\bar{\gamma} \neq 0$, in which the reference trajectories are known to be Jeffery's orbits. Given a time interval $[0, T]$, the determinant $\Delta$ of the full system is therefore an analytic function of time and, thus, cannot be identically equal to zero on $[0, T]$. Thus, the algebraic Kalman condition is satisfied, so we deduce that the linearized system is controllable. Thus, in this case, system (2.2) is locally controllable, for both position and orientation, around any reference trajectory.

Overall, the analysis of the linearized system demonstrates that we have broad local controllability of the swimmer position, except in the two particular cases noted above: namely, if the reference control $\bar{\gamma}$ is zero and either (i) $\cos\bar{\theta}_0 = 0$ (swimmer perpendicular to the shear direction) or (ii) both $\sin\bar{\theta}_0 = 0$ and $\bar{y}_0 = 0$ (swimmer moving along the midline). On the other hand, the control of the combined positional and rotational dynamics is also possible, provided the slightly more restrictive condition $\bar{\gamma} \neq 0$ holds. While this result is naturally of a local nature, it suggests that long-time global controllability can be achieved in most cases, without requiring elaborate trajectories that avoid unreachable regions.

We illustrate these results by attempting to numerically visualize the reachable space. In order to do so, we plot sample trajectories corresponding to a range of controls taken close to the reference control $\bar{\gamma}$. More precisely, let $\gamma$ be a control of small amplitude and $z$ the trajectory associated with the control $\bar{\gamma} + \gamma$. If the swimmer is locally controllable, the differences $z - \bar{z}$ for a range of different controls $\gamma$ can be expected to roughly cover a neighbourhood of the origin. Conversely, when controllability does not hold, some regions will probably appear unattainable, which provides additional information on the non-controllable directions, as featured in figure 3.

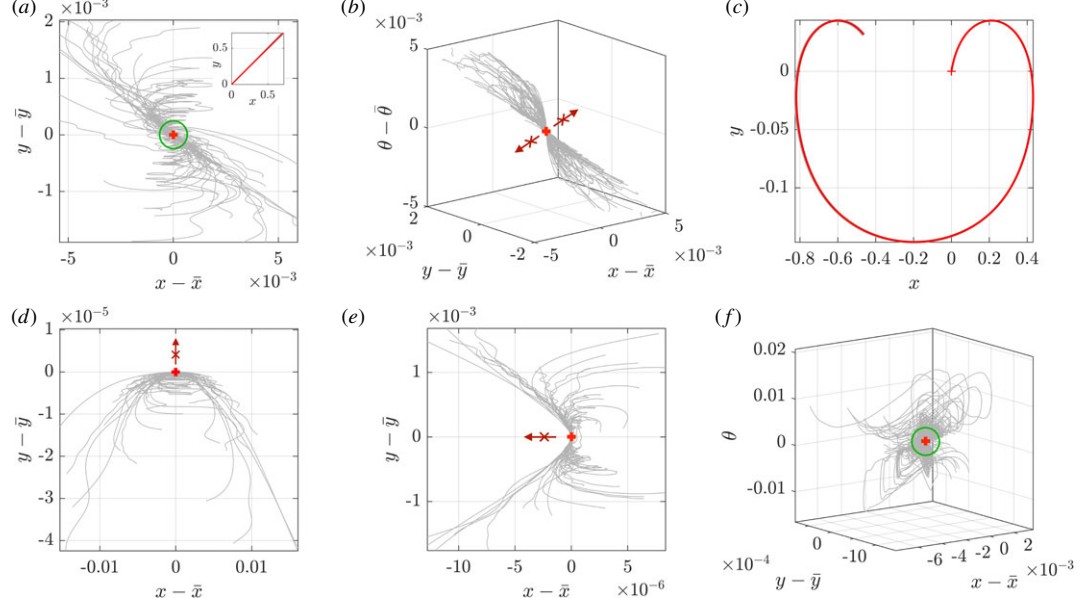

**Figure 3.** Illustration of the local controllability properties of swimmer trajectories. In panels (*a*), (*b*), (*d*) and (*e*), the components of $z - \bar{z}$ are shown for 100 controlled trajectories, with controls taken around $\bar{\gamma} = 0$ as realizations of the randomized control featured in appendix C. The reference trajectory $\bar{z}$ and the controlled trajectories are shown as red and grey curves, respectively, in the upper right insert in panel (*a*) and are approximately indistinguishable at the resolution of this figure. (*a*) A neighbourhood of the origin, indicated by a green circle, is covered in the plane $(x, y)$, but seemingly not in the full state space (*b*)—unreachable directions are highlighted by red arrows, with the initial condition here set to $z(0) = [0, 0, \pi/4]^T$. In panels (*d*) and (*e*), the initial conditions are $z(0) = [0, 0, \pi/2]^T$ and $[0, 0, 0]^T$, respectively, with the swimmer appearing to not be partially controllable for $(x, y)$ from these initial conditions, as some directions indicated by red arrows seem unreachable. In panels (*c*) and (*f*), 100 trajectories are plotted, with the reference control now $\bar{\gamma} = 20$ and the initial condition again set to $z(0) = [0, 0, \pi/4]^T$. In (*c*), the reference trajectory $\bar{z}$ is shown in red, with controlled trajectories being indistinguishable from the reference trajectory at the resolution of this figure, which underlines the local nature of our controllability results. Plotting the difference $z - \bar{z}$ between the reference trajectory and the controlled trajectories of (*c*) in (*f*), we observe that the trajectories cover a neighbourhood of the origin in the state space, indicated by a green circle.

The expression for the controls used for the simulations of figure 3 is given in appendix C. Panels (*a*) and (*b*) show that, for $\bar{\gamma} = 0$, a neighbourhood of the origin is indeed attained in the $xy$-plane, but seemingly not in the full state space $(x, y, \theta)$, in line with the above analysis. Panels (*d*) and (*e*) illustrate the lack of partial controllability that occurs in the cases (i) $\cos\theta_0 = 0$ (swimmer perpendicular to the shear direction) or (ii) both $\sin\theta_0 = 0$ and $y_0 = 0$ (swimmer parallel to the shear direction and located on the midline).

Figure 3*f* features the results for $\bar{\gamma} = 20$. In agreement with the controllability analysis, it suggests that a non-zero reference control permits the recovery of local controllability in the full state space. However, it is worth keeping in mind the local nature of this result. Indeed, the size of the neighbourhood of the origin where controllability holds may be rather small compared with the characteristic length of the system. Figure 3*c* illustrates this fact, showing that the controlled trajectories deviate little from the reference trajectory in our simulations.

Having established that the swimmer is locally controllable, a property often unexplored in contemporary studies of optimal swimmer control, let us address a more practical situation, in which we aim to reach a given target position $(x_f, y_f)$ by solving a minimal time control problem:

$$\begin{cases} \min \quad T \\ \text{s.t.} \quad \dot{z} = f(z, \gamma) \text{ on } [0, T], \\ \qquad \gamma \in [\gamma_{\min}, \gamma_{\max}], \\ \qquad z(0) = z_0, \\ \qquad x(T) = x_f, y(T) = y_f, \end{cases} \qquad (3.3)$$

where we assume $\gamma_{\min} < 0$ and $\gamma_{\max} > 0$, with the flow therefore able to be switched off.

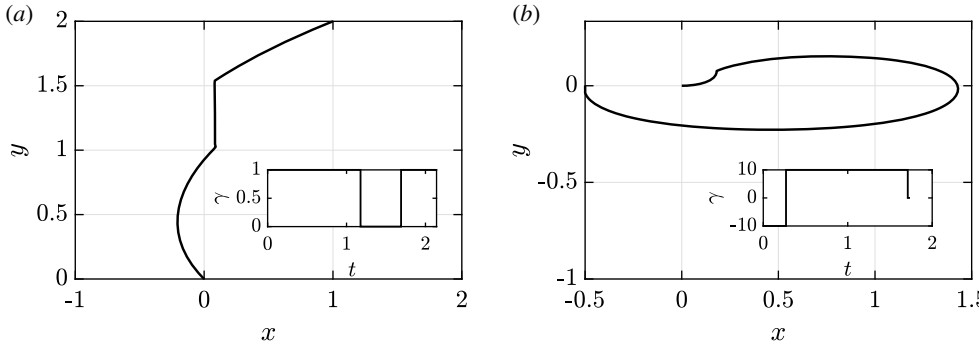

**Figure 4.** Examples of numerically computed time-optimal trajectories. (*a*) The swimmer is guided from $(x, y) = (0, 0)$ to $(x, y) = (1, 2)$, with initial orientation $\theta = 3\pi/4$. with the trajectory being a concatenation of portions of Jeffery orbits and straight lines, as predicted by the optimal control analysis. The insert showcases the bang–bang structure of the optimal control $\gamma$. (*b*) With the initial orientation now $\theta = 0$ and the target being $(x, y) = (-0.5, 0)$, which corresponds to a less favourable case according to the controllability analysis, one can see that the optimal trajectory is far from direct, requiring a much higher shear rate than in (*a*) in order to yield even this level of directness, resultant of the lack of local controllability for the swimmer's position in this case.

Let us define the adjoint state vector $\boldsymbol{p} = [p_x, p_y, p_\theta]^{\mathrm{T}}$ and write the Hamiltonian associated with (3.3):

$$H = p^0 + \boldsymbol{p} \cdot \boldsymbol{f}(\boldsymbol{z}, \gamma) = H_0 + u H_1, \tag{3.4}$$

with $p^0 \in \mathbb{R}$, $H_0 = p_x \cos\theta + p_y \sin\theta$ and $H_1 = p_x y + \frac{1}{2} p_\theta(-1 + E \cos 2\theta)$. By virtue of the Pontryagin maximum principle [52], the optimal control $\gamma$ must maximize the Hamiltonian at all times. As expected in a minimal time control problem, this shows that $\gamma$ is made of 'bang arcs' for which $\gamma$ is equal to $\gamma_{\min}$ or $\gamma_{\max}$ depending on the sign of $H_1$, and 'singular arcs' when $H_1$ vanishes on a subinterval of $[0, T]$. The value of $\gamma$ on potential singular arcs can be determined by computing $\ddot{H}_1 = 0 = \{\{H_1, H_0\}, H_0\} + \gamma\{\{H_1, H_0\}, H_1\}$, where we use Poisson bracket notation [53]. For the optimal control problem (3.3), we serendipitously have $\{\{H_1, H_0\}, H_0\} = 0$ (and $\{\{H_1, H_0\}, H_1\}$ is not identically zero), so that $\gamma$ always vanishes on the singular arcs.

Overall, $\gamma$ should follow the command law

$$\gamma(t) = \begin{cases} \gamma_{\min} & \text{if } H_1 < 0, \\ \gamma_{\max} & \text{if } H_1 > 0, \\ 0 & \text{if } H_1 = 0. \end{cases} \tag{3.5}$$

Therefore, the optimal trajectory will be a concatenation of portions of Jeffery's orbits, with $\gamma$ equal to $\gamma_{\min}$ or $\gamma_{\max}$, and line segments, with $\gamma = 0$. This is well illustrated by the optimal trajectories numerically computed in figures 4 and 5. Further, we can use the above controllability analysis to predict that some targets, those in the regions indicated by the red arrows in figure 3, will necessitate longer and less direct trajectories due to the partial lack of local controllability of the system in these cases. This is illustrated by the examples of time-optimal trajectories in figure 4: in (*a*), the target is easily reached, while the target in (*b*) is located in the locally unreachable region visible in figure 3*e*, which therefore gives rise to a longer time-optimal path, even if we permit a wider degree of control by increasing the value of $\gamma_{\max}$ in this case.

In appendix D, we further prescribe the final orientation of the swimmer, noting that the optimal control is once again given by equation (3.5), and explore the broad range of computed optima as a function of this final orientation.

# 4. Practical robustness of flow control

With partial controllability results established, evaluating the robustness of swimmer behaviour to error or uncertainty in measurements remains a necessary precursor to designing and using flow control in practice. However, any precise definition of acceptable variability or robustness is naturally context dependent. For instance, in the study of a particular swimmer, variability may be naturally measured relative to swimmer morphology, though we note that the length scale of the swimmer geometry does not enter into the dynamical system of equation (2.2). Instead, the dynamical system makes use of the

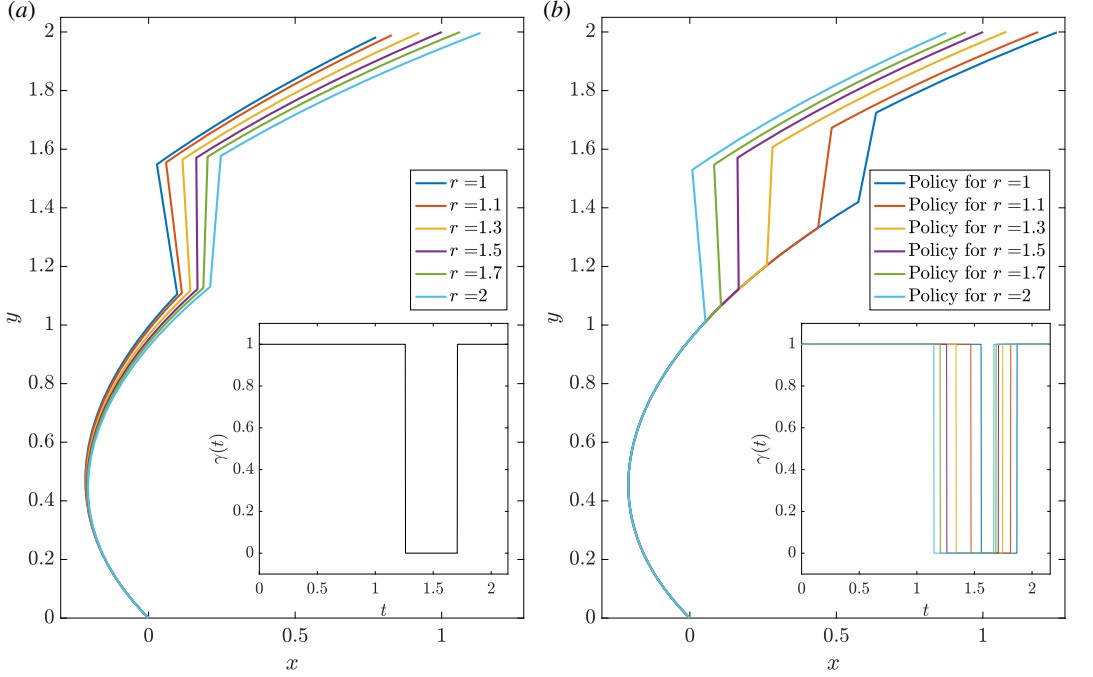

**Figure 5.** Evaluating robustness of a simple control problem. (a) Fixing a flow-control policy tailored to a swimmer with $r = 1.5$ (shown inset), we show the trajectories of swimmers with shapes $r \in [1, 2]$, each starting from the origin with fixed initial orientation. We observe qualitative similarity in the trajectory endpoints and swimmer paths, suggesting that one can conclude robustness of overall behaviour to uncertainty in swimmer shape. (b) Now tailoring flow-control policies to swimmers of various shapes (shown inset), we compute the trajectories of a swimmer of fixed shape $r = 1.5$ subject to this range of policies. In contrast to (a), now apparent are significant qualitative differences between the paths, with flow policies designed for different swimmers yielding greatly altered swimmer behaviour and, thus, demonstrating a lack of robustness and the presence of marked sensitivity to measurement error in the trajectories under different policies. Hence, the consideration of a single panel in isolation may lead to unreliable assessments of robustness, with thorough investigation that varies both shape and policy therefore warranted.

advective length scale, arising from a chosen velocity scale and time scale, which is not constrained to be equal to the length scale of the swimmer geometry. Thus, the dynamics that we will explore below are independent of the size of the swimmer, subject to the aforementioned constraints of inertia-free swimming, the validity of the spheroid approximation, and the subdominance of translational noise to the swimming dynamics that is commonplace amongst well-studied microswimmers. Hence, in what follows, we will make use of the length of swimmer trajectories, which are relative to the advective length scale, as a reference for qualitative assessments of robustness in this dimensionless setting. We direct the interested reader to appendix A for a dimensional exploration of robustness in a particular organism, *Leishmania mexicana*.

## 4.1. Evaluating robustness necessitates care

As a first concrete example, consider again the task of guiding a swimmer from $(x, y) = (0, 0)$ to $(x, y) = (1, 2)$, as in figure 4, where the initial orientation of the swimmer is fixed and a background shear flow may be prescribed as detailed in §2. For a given swimmer shape, here encoded in the aspect ratio $r$, one may construct a policy of flow control that effectively realizes the swimmer target position. Here, for illustration, we minimize travel time in line with the controllability analysis, having taken $\gamma_{min} = -1$ and $\gamma_{max} = 1$. An example of such policy is shown inset in figure 5a, tailored to a swimmer with aspect ratio $r = 1.5$. In the same figure, we also show the trajectories in the $xy$-plane for a range of swimmer shapes under this policy, sampling $r \in [1, 2]$. The trajectory for $r = 1.5$, for which the policy was designed, can be seen to effect the desired movement of the swimmer. The other curves in figure 5a showcase the variability in both path and outcome were there to be uncertainty in the measured swimmer shape, which appears to demonstrate that such differences have minimal qualitative and quantitative effects on the outcomes of the policy. From this evaluation alone, one

may be tempted to conclude that the guidance of a swimmer to (1, 2) from the origin is robust to measurement error in this set-up, with little sensitivity of the trajectory to such error.

Having fixed a policy and varied the swimmer shape, we now perform the opposite investigation, considering the effects of a variety of tailored policies on a single fixed swimmer shape. This scenario can be thought of as designing a flow policy for inaccurately measured swimmer geometry, complementary to the previous analysis, which applied a fixed policy to swimmers of various shapes. In figure 5b, we show the paths of a swimmer of fixed aspect ratio $r = 1.5$ under policies of flow control designed for swimmers with shapes of $r \in [1, 2]$. These policies are again shown inset, each being qualitatively similar though with the precise timings of flow switching differing between policies. In contrast to figure 5a, from which some notion of robustness seems apparent, here we see more-significant qualitative differences in the paths taken by the swimmer under the different policies, naturally accompanied by large quantitative distinctions. In particular, relatively large variation is present between trajectories around $y \approx 1.4$, in stark contrast to those of figure 5a. Thus, from consideration of only this plot, one is drawn to conclude that, relatively, there is a distinct sensitivity and a lack of robustness in swimmer paths to measurement errors in the swimmer shape.

Thus, while both approaches may *a priori* be thought of as sufficient explorations to establish some practical notion of robustness or sensitivity, we have seen in this simple example that the conclusions drawn from each of them may be at odds with one another, at least when considering the entirety of the swimmer trajectory. With each of these assessments therefore being incomplete and potentially misleading in isolation, this highlights that careful and thorough examination of robustness is warranted in order to draw reliable conclusions. Indeed, with this exemplar representing a very simple short-time guidance problem, this further suggests that careful examination of sensitivity from multiple viewpoints is required when considering flow-mediated swimmer control in more generality.

## 4.2. Non-monotonic responses to variation

The previous example explored a simple control scenario, demonstrating that care is necessitated when attempting to evaluate robustness of control to variations in swimmer shape. However, despite exhibiting relatively large differences in swimmer paths in some cases, the relations between applied policy and swimmer shape in figure 5 appeared simplistic, with policies that switched off the background flow at earlier times resulting in shorter swimmer trajectories, for example. One might therefore expect that such simple rules might be applicable in more generality.

We will explore this notion via a more complex example, one in which the background flow is prescribed such that a spherical swimmer, of shape $r = 1$, undergoes a closed and symmetric trajectory, as shown in figure 6a in grey. For reference, the initial condition of the swimmer and the accompanying flow policy required to elicit this behaviour are given in appendix E and are fixed throughout. In order to explore the effect of swimmer shape on swimming path, we examine the behaviours of swimmers with different shapes from this initial example, with the trajectories shown as coloured curves in figure 6b for $r \in [1, 2]$, where the motion of each swimmer has been simulated over the interval $t \in [0, 5\pi]$. Evident is a qualitative similarity in the overall shape of the swimmer paths, though the symmetry of the $r = 1$ trajectory appears broken to varying degrees for $r > 1$. Isolating a single trajectory, now shown in black, in figure 6c we see that the swimmer with $r = 1.1$ has approximately followed the path of the spherical example, though has overshot the target endpoint by some considerable margin due to increased initial rotation in the background flow compared with the spherical counterpart. Extrapolating from this minimal comparison, reassured by the simple relation between shape and trajectory seen in figure 5a, one might expect that further elongating the swimmer may yield successively greater overshoots.

However, consideration of another elongated swimmer in figure 6d, with $r = 1.7$, reveals a different narrative, with the trajectory significantly departing from the symmetry of the spherical base case and substantially undershooting the target endpoint. Here, the fast initial rotation of the swimmer has resulted in a path that only briefly enters $y > 0$ in the first half of the motion. In later portions of the motion, the non-spherical swimmer is oriented approximately 30° away from the spherical body (not shown), yielding the large difference in final position. Thus, with this undershoot not compatible with the tentative observations drawn from figure 6c, this simple relation between shape and dynamics in flow does not hold. Indeed, the lack of monotonicity in the observed relationship between swimmer geometry and the flow precludes any such simple rule for predicting, in generality, the manifestations of geometric variation or uncertainty in the observed behaviours of flow-guided swimmers. We remark, however, that simple quantifications may exist for certain features or subsets of the dynamics. In particular, a pertinent example of this is the evolution of the swimmer orientation $\theta$, which, if the

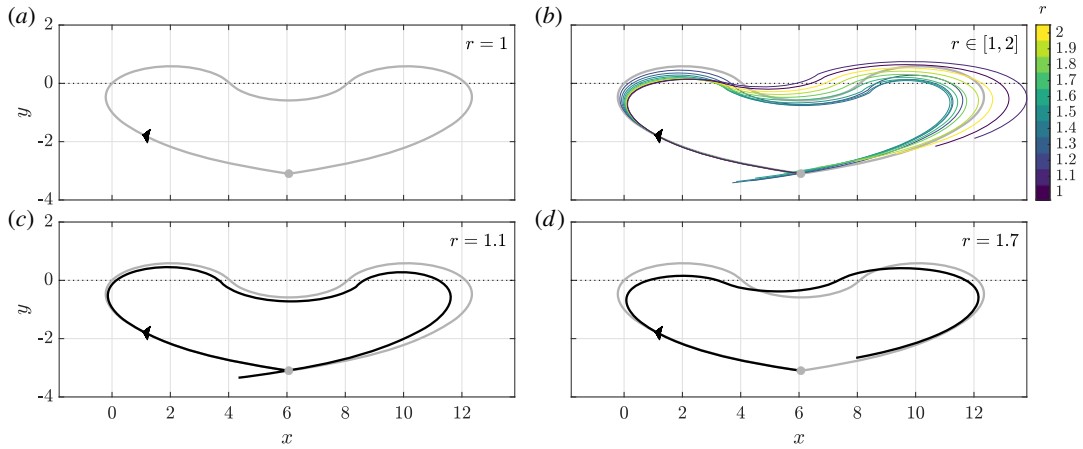

**Figure 6.** Complex, non-monotonic responses of swimmer behaviour to changes in shape. Prescribing an initial condition and fixed flow policy so as to elicit the symmetric closed trajectory in the $xy$-plane shown in (*a*) for a spherical swimmer ($r = 1$), we simulate the motion of a range of swimmers in this background flow, with $r \in [1, 2]$, with these trajectories shown as coloured curves in (*b*). Panels (*c*) and (*d*) highlight single trajectories in comparison with the symmetric target path, the latter being shown as a grey curve throughout the figure. (*c*) For $r = 1.1$, the swimmer overshoots the desired endpoint. (*d*) Increasing $r$ further to 1.7 yields an undershoot in final position, thereby highlighting the non-monotonic response of a swimmer behaviour to changes in shape. The black arrowhead shows the direction of traversal of the swimmers around their trajectories, with the initial position shown as a filled grey circle.

swimmer is subject to constant shear flow, follows a canonical Jeffery's orbit with a period that is monotonically increasing with respect to swimmer eccentricity, while the corresponding trajectory in the $xy$-plane may nevertheless be intricate and defy similar analysis.

## 4.3. Flow-amplification of diffusive effects

Thus far we have considered the trajectories of swimmers in the absence of any diffusive effects, though many potential biological swimmers, such as motile bacteria, warrant the inclusion of such rotational noise. The effects of translational noise, however, are comparatively small for many self-propelled swimmers, with the corresponding Péclet number being sufficiently high so as to justify their frequent neglect [44]. We therefore modify the orientation dynamics, replacing the ordinary differential equation of equation (2.2) for $\theta$ with the stochastic differential equation

$$\mathrm{d}\theta_t = \frac{1}{2}\gamma(t)(-1 + E\cos 2\theta_t)\,\mathrm{d}t + \sqrt{D}\,\mathrm{d}W_t, \tag{4.1}$$

where $W$ denotes a Wiener process and $D$ is a diffusion coefficient that will characterize the contribution of rotational noise. Here, $D$ is constant, therefore there is no distinction between treating equation (4.1) as an Ito or Stratonovich stochastic differential equation. Thus, solution via the Ito algorithm of the Euler–Maruyama method, as used here, is legitimate even though Stratonovich calculus is often advocated for physical systems [54].

In order to investigate the interactions between a guiding flow and rotational diffusion, we consider the dynamics of a spherical swimmer, taking $r = 1$, with and without a background flow that is prescribed so as to yield a oscillatory trajectory, as shown in figure 7a. This flow is given by

$$\gamma(t) = \begin{cases} +1, & 2n\pi \leq t < (2n+1)\pi, \\ -1, & (2n+1)\pi \leq t < (2n+2)\pi, \end{cases} \quad n \in \mathbb{Z}, \tag{4.2}$$

where we simulate the motion for $t \in [0, 5\pi]$ from the initial condition $(x(0), y(0), \theta(0)) = (0, 0, \pi/4)$. In particular, such a flow is a pertinent exemplar in the consideration of swimmer behaviour and guidance, with consecutive bang arcs featuring in solutions to minimal time control problems, as highlighted by equation (3.5).

In figure 7a, we observe that the swimmer's trajectories in the absence of noise are indeed the expected oscillatory and straight paths, corresponding to the cases with and without the guiding flow, respectively. Introducing diffusive effects, in figure 7b we note the progressive divergence of the swimmer paths away from the noise-free case, with these effects clearly evident in figure 7c and d.

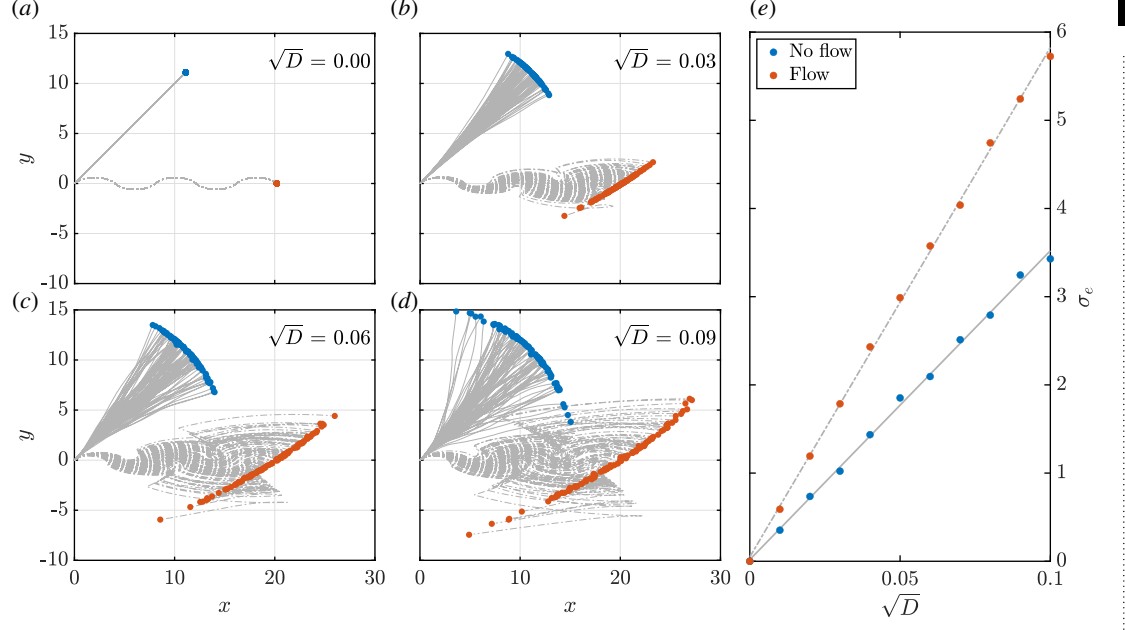

**Figure 7.** Amplification of rotational noise by flow. With the diffusion parameter $D$ fixed in each of the panels (a)–(d), we simulate the motion of a spherical swimmer from a prescribed initial condition for $t \in [0, 5\pi]$, prescribing either no flow or the background flow given in equation (4.2), with trajectories shown as solid and dot-dashed lines in each of these cases, respectively. With $D = 0$ in (a), we recover noise-free trajectories, straight in the absence of flow and oscillatory when the guiding flow is present. Increasing the level of noise, we observe an expected increase in the spread of swimmer trajectories across panels (b)–(d), with the endpoints of trajectories shown as blue and red dots for the no-flow and with-flow cases, respectively. Noting a somewhat wider spread of swimmers in the presence of background flow, particularly evident in (d), we plot a measure of the spread of the final swimmer position, defined in equation (4.3) and denoted by $\sigma_e$, for a range of values of the diffusion parameter in (e). Now quantitatively, we observe a marked increase in the effective diffusion of the swimmer when the background flow is present, with the flow amplifying diffusive effects by approximately 65% in this case. In panels (a–d), 100 realizations of the swimmer motion are presented, while the quantitative measure $\sigma_e$ of (e) is computed using 1000 realizations in each case.

Indeed, visible in these latter panels is a wider spread of the final swimmer positions when the flow is present compared with motion in the absence of flow. In order to assess this quantitatively, in the separate cases of motion with and without flow we compute a measure of the spread of the final positions of the swimmers for a fixed $D$, defined explicitly as

$$\sigma_e = \sqrt{\mathrm{Var}[x_e] + \mathrm{Var}[y_e]}, \tag{4.3}$$

where the subscripts denote evaluation at the end of the path. Simulating 1000 trajectories for a range of considered $D$, this measure is plotted in figure 7e, from which a clear distinction between the cases of flow and no flow is evident. In agreement with qualitative assessments of figure 7a–d, the effect of the flow is tantamount to increasing the diffusion coefficient of the swimmer, in this case amplifying diffusive effects by approximately 65%. Of note, though not shown, this conclusion is qualitatively unchanged when considering motion away from the axis of the shear at $y = 0$, as well as when considering a flow with constant shear rate.

Therefore, while the inclusion of rotational diffusion might be expected to negatively impact on robustness, we have observed that the presence of a background guiding flow in this case enhances the effects of this noise, with robustness therefore necessitating an evaluation that includes the details of the controlling flow and, hence, is unable to be extrapolated from swimming behaviours in a quiescent environment.

# 5. Discussion

In this work, we have considered a simply posed problem of guiding a model microswimmer with a modifiable shear flow, examining various desirable control properties of the associated dynamical system via both theoretical and numerical explorations. By direct simulation, we have seen that

uncertainty in swimmer morphology can lead to varying behaviours under the same imposed flow conditions, concluding in particular that the evaluation of practical robustness necessitates thorough analysis to afford confidence in a particular regime of control. This requirement of detailed context-dependent investigation into the robustness of swimmer control was further compounded by the existence of complex non-monotonic responses to simple variations in swimmer geometry, precluding the existence of simple rules of robustness in generality. Having quantified idealized swimmer morphology by a single parameter in this study, in appendix A we cast this analysis in a practical context, noting a robustness of control to variations in flow response time that may be associated with contemporary microfluidic devices. More generally, while the extension of this investigation to swimmers with more intricate geometry represents a pertinent direction for future work, the cautionary notes of this work serve to exemplify the care that will probably be needed in more complex settings, with perhaps surprising degrees of sensitivity to uncertainty, both low and high, found in the explorations presented in this study.

Despite the identified practical concerns, which we note may be significantly mitigated by confidence in swimmer morphology and flow control, theoretical analysis has demonstrated that flow control represents a plausible method for realizing microswimmer guidance, having established the local controllability of the positional and rotational dynamics of the swimmer, with a notable exception when the reference flow is zero. In that latter case, numerical investigation of the full nonlinear system reveals apparently inaccessible regions of the state space. With focus therefore drawn to controlling swimmer position, which nonetheless could represent a feat of significant utility in practice, we addressed the minimal time control problem, establishing that the optimally chosen flow rate is either extremal or zero. This simple result provides helpful guidance for optimal control design in this context and illustrates the general principle of optimizing flow-mediated control.

Lastly, we considered the effects of rotational noise, which is of pertinence to small-scale helical swimmers such as bacteria. In particular, augmenting our simple control system with unbiased rotational noise, we quantitatively assessed the interactions between flow and diffusive effects on swimmer position, prescribing a policy of shear control. Computing a length scale of effective dispersion, we observed that the background flow acts to increase the effects of diffusive noise, resulting in a marked increase in the variability of swimmer location in this case. Therefore, we are led to conclude that assessments of the impacts of noise on swimming in the absence of flow cannot naively be extrapolated to cases where flow is present. As such, careful evaluation of diffusive effects is warranted in general, with the potential for complex interplay between multiple aspects of the control system. Though we have only considered prescribed controls in this work, addressing the impacts of noise may also warrant the use of control-feedback loops, in which the control is adjusted in real time to account for the measured impacts of noise or other complicating factors [6,55].

The evaluations and assessments presented in this work have been naturally restricted to a simple framework. However, the implications of the findings apply more generally as cautionary examples, highlighting a lack of universal robustness that has been without thorough investigation. In practice, the neglect of translational Brownian noise and inertial effects in this work limits the presented analysis to swimmers on the microscale, with, in general, larger swimmers experiencing inertial effects and smaller swimmers being significantly impacted by translational noise. Despite these restrictions, this class of swimmers is broad, including canonically studied examples such as motile bacteria and spermatozoa. This work may be extended to consider more complex systems, for example, multiswimmer environments and confined domains, with the latter already having been associated with effecting long-time changes in the behaviour of swimmers in shear [16], though faithful study is likely to require the inclusion of additional factors such as steric forces. More generally, application of the Faxén relation, summarized in appendix B, will enable the ready inclusion of sophisticated flows in future study. Further work may also extend to considerations of three-dimensional motion, with the local controllability analysis being readily generalizable to such broader contexts. Indeed, with established robustness and feasibility being a desirable, if not necessary, precursor to practical applications of swimmer guidance, such an analysis could be extended to other modalities of microswimmer control, with simple acoustically guided swimmers expected to give rise to a similarly tractable control system.

In summary, we have posed a simple control system for flow-mediated swimmer guidance, establishing the theoretical controllability of the planar positional dynamics of a spheroidal swimmer in a shear flow. Numerical explorations of the model dynamics have revealed a surprising sensitivity of swimmer motion in flow to uncertainty, with the effects of rotational diffusion in particular being amplified by the presence of flow. Nevertheless, using both practical and theoretical evaluations of

control and controllability, this analysis has suggested the potential efficacy of a widely applicable mechanism of swimmer guidance, while highlighting the importance of thorough assessments of robustness, control and controllability.

Data accessibility. The data used and generated in this paper is freely available from the University of Oxford Research Archive (ORA) via the link http://dx.doi.org/10.5287/bodleian:wr6d1YEVP.

Authors' contributions. C.M. carried out the controllability analysis, drafted and critically revised the manuscript, and designed the study. K.I. and E.A.G. designed the study and critically revised the manuscript. B.J.W. carried out the robustness analysis, drafted and critically revised the manuscript, and designed the study.

Competing interests. At the time of writing, Kenta Ishimoto is a Board Member of Royal Society Open Science, but had no involvement in the review or assessment of the paper.

Funding. C.M. is a JSPS International Research Fellow (PE20021). K.I. acknowledges JSPS-KAKENHI for Young Researchers (grant no. 18K13456) and JST, PRESTO, Japan (grant no. JPMJPR1921). C.M. and K.I. were partially supported by the Research Institute for Mathematical Sciences, an International Joint Usage/Research Center located at Kyoto University. B.J.W. is supported by the UK Engineering and Physical Sciences Research Council (EPSRC), grant no. EP/N509711/1.

# Appendix A. Controlling *Leishmania mexicana* and the impacts of flow response times

As a particular example, we consider the control of a model *Leishmania mexicana* cell, with Walker *et al.* [16] reporting an effective aspect ratio of $r = 5.49$. We will also make use of typical *Leishmania mexicana* length and velocity scales, taking the cell body length to be 10 μm and the propulsive velocity to be 20 μm s$^{-1}$ [56]. Of particular note, the computation of this motion is performed by simulating the general dimensionless system (2.2), with quantities subsequently rescaled to reflect the *Leishmania mexicana* velocity scale. Here, we redimensionalize with a velocity scale of 20 μm s$^{-1}$ that, subject to a choice of time scale of 1 s, gives rise to a natural advective length scale of 20 μm. We note that this need not correspond to the length scale of the swimmer geometry, as the dimensionless dynamical system of equation (2.2) is independent of our choice of length scale given a fixed velocity scale, up to rescalings in time. In order to illustrate the effects of the response times of flows in practice on swimmer control, we specify three similar flow controls, each of which seeks to transition between specified shear rates, which we constrain to be at most 20 s$^{-1}$. These controls differ in the time taken for each to achieve the intended shearing rates after it is prescribed, with the sharpest control switching between shear rates within 1 ms and the least responsive control transitioning over a period of 50 ms. These piecewise linear controls are illustrated in figure 8*b*, with flow switching initiated at the points marked with red dots. Shown in green is the least responsive control, which we see indeed transitions most gradually, in contrast to the near-instantaneous switching of the blue and yellow curves, which correspond to response times of 1 and 10 ms, respectively.

The corresponding paths, as well as the caricatured morphology of *Leishmania mexicana*, are shown in figure 8*a*, from which we see a significant deviation of the swimmer from its otherwise straight rightward path due to the flow. Perhaps surprisingly, the range of considered flow response times realize approximately the same swimmer location at the end of the motion, suggesting that the impacts of practical flow response times on the feasibility of control are relatively minor, in agreement with the observations of Walker *et al.* [16]. Of particular note, however, is that flow response times may be readily incorporated into any numerical optimization or control design problems, so that swimmer control may be reliably modelled and realized even if switching times are not near-instantaneous, unlike those of §§3 and 4.

# Appendix B. Deriving the equations of motion

The linear velocity $\boldsymbol{U}$ and angular velocity $\boldsymbol{\omega}$ of a force and torque-free prolate spheroid with unit major axis in a general flow $\boldsymbol{u}^{\infty}$ are given by the Faxén Laws, which, following Kim & Karrila [43], may be stated as

$$\boldsymbol{U} = \frac{1}{2e} \int_{-e}^{e} \left[ 1 + (e^2 - \xi^2) \frac{1 - e^2}{4e^2} \nabla^2 \right] \boldsymbol{u}^{\infty}(\boldsymbol{x}) \, \mathrm{d}\xi \tag{B 1}$$

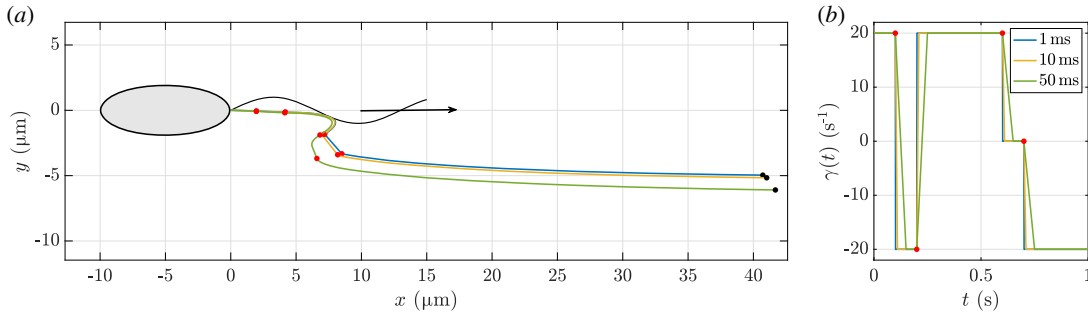

**Figure 8.** A model flagellated swimmer, *Leishmania mexicana*, and its control by flow. (*a*) We show a caricatured swimmer with typical body length of 10 µm, which would swim in approximately the direction of the black arrow in the absence of external flow. Shown in colour are the trajectories of the swimmer under various flow controls, with red dots corresponding to the times at which the shear rate of the flow is changed. Black dots highlight the position of the swimmer after 1 s. (*b*) The controls used to effect the trajectories shown in (*a*), each corresponding to different flow response times, which are given in the legend. The shear rate begins to switch at the times indicated by the red dots, with the flows responding over times ranging from 1 to 50 ms. Irrespective of the response time of the set-up, the background flow is able to enact significant change in the swimmer path, with all trajectories ending in qualitatively similar locations after 1 s. All quantities are dimensional in this figure.

and

$$\boldsymbol{\omega} = \frac{3}{8\,e^3} \int_{-e}^{e} (e^2 - \xi^2) \nabla \wedge \boldsymbol{u}^{\infty}(\boldsymbol{x}) \mathrm{d}\xi$$
$$+ \frac{3}{4e(2 - e^2)} \int_{-e}^{e} \left[ (e^2 - \xi^2) \left( 1 + (e^2 - \xi^2) \frac{1 - e^2}{8\,e^2} \nabla^2 \right) \right] \boldsymbol{d} \wedge (\boldsymbol{e}^{\infty}(\boldsymbol{x}) \cdot \boldsymbol{d}) \, \mathrm{d}\xi. \tag{B 2}$$

Here, the unit vector $\boldsymbol{d}$ points along the long axis of the ellipsoid, making an angle $\theta$ to the *x*-axis of the laboratory frame, $\boldsymbol{e}^{\infty} = [\nabla \boldsymbol{u}^{\infty} + (\nabla \boldsymbol{u}^{\infty})^{\mathrm{T}}]/2$ is the rate of strain tensor of the background flow, and $e = \sqrt{1 - 1/r^2}$ is the eccentricity of the spheroid. The variable of integration $\xi$ parametrizes the major axis of the ellipsoid, with the position $\boldsymbol{x}$ on this axis being given explicitly by

$$\boldsymbol{x}(\xi) = \boldsymbol{x}_c + \xi \boldsymbol{d}, \quad \xi \in [-e, e], \tag{B 3}$$

where $\boldsymbol{x}_c$ is the location of the centre of the ellipsoid. In the case of planar motion, we may identify $\boldsymbol{x}_c = x\boldsymbol{e}_x + y\boldsymbol{e}_y$, as in §2.

These equations can be applied to general background flows, generating a control system akin to that of equation (2.2), though with complexity naturally inherited from the form of the background flow. For the case of the linear shear flow considered in this work, as given in equation (2.1), these integrals can be readily computed by hand, yielding

$$\boldsymbol{U} = \gamma(t)(\boldsymbol{x}_c \cdot \boldsymbol{e}_y)\boldsymbol{e}_x = \gamma(t)y\boldsymbol{e}_x, \quad \boldsymbol{\omega} = \frac{1}{2}\gamma(t)\left( \frac{e^2}{2 - e^2} \cos 2\theta - 1 \right) \boldsymbol{e}_z, \tag{B 4}$$

having written $\boldsymbol{d} = \cos\theta \boldsymbol{e}_x + \sin\theta \boldsymbol{e}_y$ and defined the out-of-plane unit vector $\boldsymbol{e}_z$ to complete the right-handed orthonormal basis $\{\boldsymbol{e}_x, \boldsymbol{e}_y, \boldsymbol{e}_z\}$. Differentiating this expression for $\boldsymbol{d}$ with respect to time and applying the standard relation $\dot{\boldsymbol{d}} = \boldsymbol{\omega} \wedge \boldsymbol{d}$, we arrive at the angular evolution equation

$$\dot{\theta} = \frac{1}{2}\gamma(t)\left( \frac{e^2}{2 - e^2} \cos 2\theta - 1 \right). \tag{B 5}$$

This reduces to the expression for the angular component of the swimmer motion in equation (2.2), noting that the shape parameter $E$ may be written in terms of the eccentricity $e$ via $E = e^2/(2 - e^2)$. By the linearity of Stokes equations, the velocity of the swimmer due to self-propulsion may simply be added to $\boldsymbol{U}$, yielding precisely the control system of equation (2.2), noting that the direction of self-propulsion is given by $\boldsymbol{d}$.

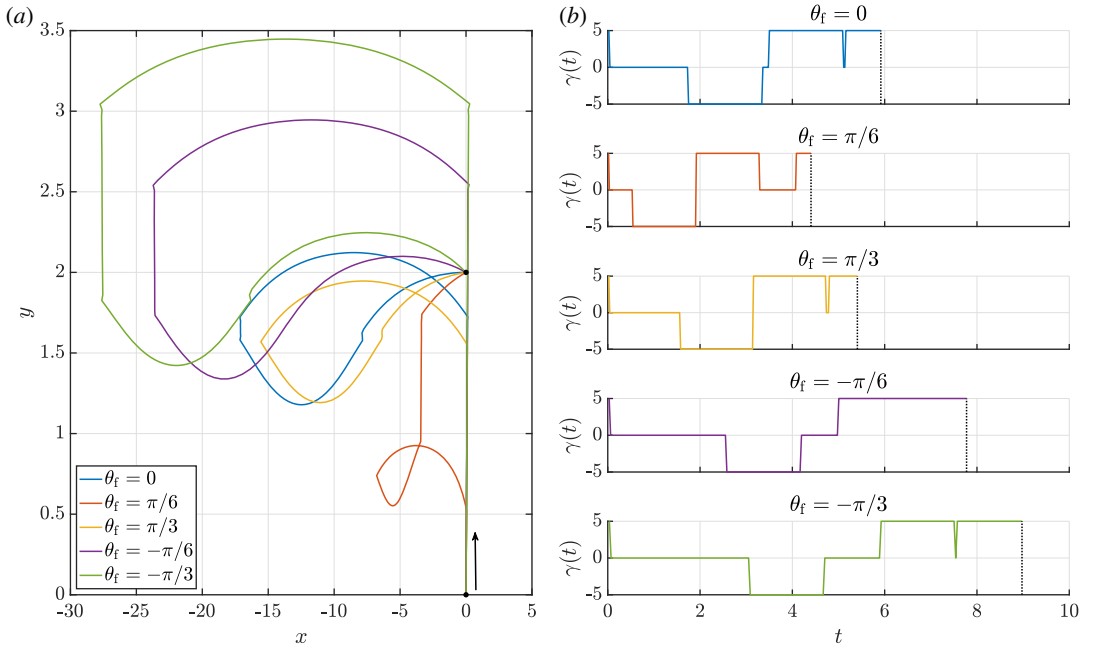

**Figure 9.** Prescribing final orientation in an optimal-time problem. Optimal trajectories (*a*) and the corresponding controls (*b*) for minimizing the travel time between the origin and the point (1, 2), each shown as black dots in (*a*). Here, with an initial swimmer orientation of $\pi/2$, indicated by the black arrow in (*a*), we explore the effects of prescribing various final orientations $\theta_f$, noting large variation in both the character of the resulting trajectories and their length. Correspondingly, indicated as vertical dashed lines in (*b*), the minimal times vary significantly with $\theta_f$. Axes in (*a*) are shown stretched for visual clarity and we have constrained $\gamma(t)$ to lie between $\pm 5$ s$^{-1}$.

# Appendix C. Flow controls for §3

The controls $\gamma(t)$ used for the numerical simulations of figure 3 are realizations of the control

$$\gamma(t) = \bar{\gamma} + \frac{1}{30}(\gamma_1 \sin(10\omega_1 t + \phi_1) + \gamma_2 \sin(100\omega_2 t + \phi_2)),$$

where $\bar{\gamma} = 0$ in panels (*a*), (*b*), (*d*) and (*e*), $\bar{\gamma} = 20$ in panels (*c*) and (*f*), and $\gamma_1, \gamma_2, \omega_1, \omega_2, \phi_1$ and $\phi_2$ are drawn uniformly at random from $[-1, 1]$.

# Appendix D. Controlling rotational dynamics

Despite being unable to establish as broad a local controllability result for the full system given in equation (2.2), in contrast to the reduced (*x*, *y*) system, the analogue of the time minimization problem equation (3.3) with additional restrictions on $\theta$ may still be explored with sufficiently large bounds on the control $\gamma$. This augmented problem naturally inherits the potential sensitivities explored in §4. and retains the bang–bang form of the optimal controls. Through exemplar computation, shown in figure 9, we see highlighted the significance of rotational restrictions on feasible swimmer trajectories, with great qualitative variation exhibited by the computed trajectories as a function of prescribed eventual orientation. Further, the computed controls, shown in figure 9*b*, highlight the larger variation in time taken to reach the endpoint of the trajectory, which here has coordinates (*x*, *y*) = (1, 2). Thus, in practice, the goal of controlling the final orientation of the swimmer may need to be carefully prioritized alongside other objectives, such as minimizing the time or energy required to realize an identified control.

# Appendix E. Set-up and flow policy for §4.2.

The initial condition and flow policy corresponding to §4.2. are given by

$$(x(0), y(0), \theta(0)) = (6.0635, -3.0989, 2.5725), \quad \gamma(t) = \begin{cases} +1, & t - t^{\star} < \pi, \\ -1, & \pi \le t - t^{\star} < 2\pi, \\ +1, & 2\pi \le t - t^{\star}. \end{cases} \quad \text{(E 1)}$$

Here $t^{\star} = 3.5742$ and $t \in [0, 3\pi + 2t^{\star}]$, yielding an approximately closed trajectory.

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
