## [Peer Review File · Royal Society Open Science]

Review History

Decision letter (RSOS-211141.R0)

Dear Dr Walker:

I am pleased to inform you that your manuscript entitled "Control and controllability of microswimmers by a shearing flow" is now accepted for publication in Royal Society Open Science.

Please ensure that you send to the editorial office an editable version of your accepted manuscript, and individual files for each figure and table included in your manuscript. You can send these in a zip folder if more convenient. Failure to provide these files may delay the processing of your proof.

You can expect to receive a proof of your article in the near future. Please contact the editorial office (openscience@royalsociety.org) and the production office (openscience_proofs@royalsociety.org) to let us know if you are likely to be away from e-mail contact – if you are going to be away, please nominate a co-author (if available) to manage the proofing process, and ensure they are copied into your email to the journal. Due to rapid publication and an extremely tight schedule, if comments are not received, your paper may experience a delay in publication.

on behalf of Professor Rachel Bearon (Associate Editor) and Professor Mark Chaplain (Subject Editor).
